# Atropine-Phosphotungestate Polymeric-Based Metal Oxide Nanoparticles for Potentiometric Detection in Pharmaceutical Dosage Forms

**DOI:** 10.3390/nano12132313

**Published:** 2022-07-05

**Authors:** Seham S. Alterary, Maha F. El-Tohamy, Gamal A. E. Mostafa, Haitham Alrabiah

**Affiliations:** 1Department of Chemistry, College of Science, King Saud University, P.O. Box 22452, Riyadh 11451, Saudi Arabia; salterary@ksu.edu.sa (S.S.A.); moraby@ksu.edu.sa (M.F.E.-T.); 2Department of Pharmaceutical Chemistry, College of Pharmacy, King Saud University, P.O. Box 2457, Riyadh 11451, Saudi Arabia

**Keywords:** atropine sulfate, metal oxides (ZnO; MgO), nanoparticles, polymeric sensors, commercial dosage forms

## Abstract

The new research presents highly conductive polymeric membranes with a large surface area to volume ratio of metal oxide nanoparticles that were used to determine atropine sulfate (AT) in commercial dosage forms. In sensing and biosensing applications, the nanomaterials zinc oxide (ZnONPs) and magnesium oxide (MgONPs) were employed as boosting potential electroactive materials. The electroactive atropine phosphotungstate (AT-PT) was created by combining atropine sulfate and phosphotungstic acid (PTA) and mixing it with polymeric polyvinyl chloride (PVC) with the plasticizer o-nitrophenyl octyl ether (*o*-NPOE). The modified sensors AT-PT-ZnONPs or AT-PT-MgONPs showed excellent selectivity and sensitivity for the measurements of atropine with a linear concentration range of 6.0 × 10^−8^ − 1.0 × 10^−3^ and 8.0 × 10^−8^ − 1.0 × 10^−3^ mol L^−1^ with regression equations of E_(mV)_ = (56 ± 0.5) log [AT] − 294 and E_(mV)_ = (54 ± 0.5) log [AT] − 422 for AT-PT-NPs or AT-PT-MgONPs sensors, respectively. The AT-PT coated wire sensor, on the other hand, showed a Nernstian response at 4.0 × 10^−6^ − 1.0 × 10^−3^ mol L^−1^ and a regression equation E_(mV)_ = (52.1 ± 0.2) log [AT] + 198. The methodology-recommended guidelines were used to validate the suggested modified potentiometric systems against various criteria.

## 1. Introduction

Advances in nanoscience and nanotechnologies are becoming increasingly significant in the design and engineering of sensors and biosensors. Metal and metal oxide nanoparticles have been used to improve the sensitivity, electrical conductivity, and performance of sensors. Many new signal transduction technologies in sensors and biosensors have been facilitated because of the usage of these nanomaterials. Nanosensors, probes, and other nanosystems have enabled easy and speedy in vivo studies due to their submicron dimensions [1]. Metal oxide nanoparticles are high-activity nanostructures that can be synthesized and employed in various applications. Their potential is so vast that they may be used in a variety of sensing and biosensing approaches [2] due to their growing demand and rapid investigation into sensor fabrication; indeed, they have emerged as feasible solutions for overcoming the drawbacks of microstructures. Furthermore, the unique structures and features of these metal oxides have been recognized as a vital key in the development of a large range of electronics [3,4], cancer diagnosis and treatment [5,6], and immunosensing approaches [7,8].

Polymeric sensors functionalized with metal oxide nanoparticles have a number of benefits over typical polymers including chemical resistance, high conductivity, biocompatibility, and flexibility [9]. Recent advances in scientific areas such as pharmaceutics [10], drug analysis [11], medicine [12], industry [13], and electronics [14] necessitate the development of novel sensing knowledge that possesses low power consumption, firmness, high selectivity, and sensitivity.

Various studies are currently being conducted on zinc oxide nanoparticles (ZnONPs) and magnesium oxide nanoparticles (MgONPs). In addition, the application of both metal oxides in a variety of sectors including immunomodulatory agents [15,16], bactericidal agents [17,18], catalysis [19,20], tissue imaging [21,22], sensors [23,24], food packaging [25,26], and chemotherapeutics [27,28] has received more attention. In sensor construction, the thickness of the coating layer and chemical contents with respect to the loading ceria were examined using various microscopic and spectroscopic approaches including scanning electron microscopy (SEM) and X-ray photoelectron spectroscopy (XPS) [29,30]. Recently, various methods have been reported for the synthesis of ZnONPs and MgONPs including thermal hydrolysis [31,32], microwave-assisted [33,34], and sol–gel seeding [35,36], and electrochemical deposition [37,38].

Metal wires with high conductivity such as aluminum, iron, platinum, gold, copper, and silver are commonly used in potentiometric wire-based sensors. The clean pure wire is coated with a polymeric cocktail that possesses the active sites [39].

Electrochemical analytical methods such as amperometry, potentiometry, and conductometry have found a number of interesting applications in various scientific areas including the analysis of drugs, biomedical applications, clinical diagnosis, and environmental studies [40,41,42]. For several biological features, potentiometric systems are more reliable and economical [43,44]. These methods are also quick in terms of how long the analysis takes [45]. To increase their sensitivity and overcome the detection limits, a number of biosensing electrodes have been functionalized with metallic oxides [46,47].

One of the most promising electrochemical methods, also referred to as a self-powered method, is the potentiometric method. When analytes build up underneath an electrostatic mechanism, a potential difference between the surfaces of the working electrode and the reference electrode forms, which is how potentiometric measurements in self-powered sensors are carried out [48].

Potentiometric sensors typically consist of membranes made from high-molecular weight polyvinyl chloride (PVC), plasticizers such as dioctyl sebacate (DOS), dioctyl phthalate (DOP), dibutyl sebacate (DBS), and dibutyl phthalate (DBP) as well as ortho-nitrophenyl octyl ether (*o*-NPE). These membranes may also be constructed from lipophilic ions or molecules, which act as active components to promote particular analyte interactions in the membrane regions, enabling the pre-detection of the sensor selectivity [49].

Metal wire with a high conductivity such as copper, iron, silver, aluminum, and platinum is frequently used to make potentiometric wire-based sensors. The substrate of the metal wire is covered with a polymeric cocktail that contains the active areas of the selective membrane. A white powder called phosphotungstic acid (PTA) may dissolve in both alcohol and water. It can also catalyze ion-exchange via the interfaces of hydrophobic membranes when paired with a suitable ion–pair complex [50].

Atropine is a tropane alkaloid found in plants of the solanaceous family including Datura, and *Belladonna atropa* has anticholinergic hallucinogenic qualities due to its antagonistic actions on acetylcholine receptors. Atropine is poisonous as a result of this impact, but its psychedelic properties make it a desirable drug [51]. Atropine is a drug that is used for a variety of purposes including the treatment of Parkinson’s disease [52,53] and the prevention of myopia loss [54]. Organophosphate poisoning can also be treated with atropine [55]. Its chemical structure is shown in Figure 1. Both medical and pharmacological uses rely on the determination of atropine. The basic methods to determine atropine are chromatographic separation methods [56,57,58]. These processes are considered as high-cost tools and necessitate the use of an extraction solvent. Furthermore, the complexity of such methods needs the use of highly trained personnel.

Low-cost equipment, on the other hand, is utilized to test atropine including spectrophotometry [59], luminescence [60], electrochemiluminescence sensors [61], biosensor [62], voltammetry [63], and potentiometry [64]. These approaches, on the other hand, necessitate more time-consuming manipulation and costly chemicals. In the meantime, potentiometric methods based on PVC sensors or electrodes have emerged as a viable alternative due to key advantages over those methods such as low expense and quick analysis time with high selectivity in a variety of matrices [65]. The reported methods for atropine are based on the use of atropine–ion pairs or ion-exchangers as electroactive materials such as atropine-reineckate [66], atropine-phosphotungstate [66], atropine-tetrakis(4-chlorophenyl)-borate [64] are used for the determination of atropine based on the exchange equilibrium. As a result, neutral ionophores such as cyclodextrin [67], phosphorated calix-arene derivatives [68], and valinomycin [69] look to be better characteristics with regard to the selectivity and linearity.

The purpose of this study was to create modified metal oxide AT-PT-ZnONPs and AT-PT-MgONPs coated wire sensors with high sensitivity and selectivity for detecting AT in commercial products. The use of modified metal oxide nanomaterials in the construction of sensors has been proposed to improve the sensitivity and selectivity of the proposed sensor. The use of nanomaterials in the construction of the sensors has been proposed to improve the sensitivity and selectivity of their potential. The incorporation of ZnONPs or MgONPs with the electroactive complex in the polymeric matrix will have a great impact on the sensitivity and selectivity of the suggested sensors for the quantification of the selected medication. Method validation was performed according to the ICH criteria [70] to establish the analytical appropriateness of the indicated systems. A comparison of the proposed modified potentiometric systems and the normally manufactured type was also carried out.

## 2. Experimental

### 2.1. Reagents and Materials

All of the materials used were of analytical quality. Sigma-Aldrich (Hamburg, Germany) provided high molecular weight PVC powder, *o*-nitrophenyloctyl ether (*o*-NPOE), tetrahydrofuran (THF) (>99%), and phosphotungstic acid. Atropine sulfate monohydrate was acquired from Fluka AG (Buchs, Switzerland). Zinc chloride and magnesium sulfate were purchased from BDH (Poole, UK). Double distilled water was used in all experiments. The atropine vial (1.0 mg mL^−1^) and eye drops (1%) were acquired from the local pharmacy. A standard stock solution of atropine (1.0 × 10^−2^ mol L^−1^) was prepared in deionized water and different working samples were prepared by serial dilution.

### 2.2. Instruments

The HANNA-pH-211 (Hanna Instruments, Smithfield, RI, USA) microprocessor pH meter was used to produce potentiometric measurements. A glass pH electrode and a double junction (Ag/AgCl reference electrode; model 90-02, Orion 81-02) were used. Various spectroscopic and microscopic techniques including a UV-10S Genesis spectrophotometer (Thermo-Fischer Scientific, Madison, WI, USA) were used to characterize the produced metal oxide nanoparticles (ZnO and MgO). A Spectrum BX spectrometer for Fourier transform infrared spectroscopy (FTIR) (PerkinElmer, Waltham, MA, USA), X-ray diffraction (XRD) (Shimadzu, Kyoto, Japan, XRD-6000 diffractometer), scanning electron microscope, and transmission electron microscope (SEM, JSM-7610F, and TEM, JEM2100F, Tokyo, Japan) were also applied for the characterization of nanoparticles. The presence of the Zn, Mg, and O elements in the produced nanomaterials was also detected by energy-dispersive X-ray spectroscopy (EDX-8100, Shimadzu, Kyoto, Japan).

### 2.3. Preparation of AT-PT Electroactive Material

The electroactive compound AT-PT was prepared by mixing three to one (75 mL of atropine to 25 mL of phosphotungstic acid) of equal aqueous solution (1.0 × 10^−2^ mol L^−1^) of both aqueous solutions. A grayish precipitate of AT-PT was formed. The precipitate was rinsed several times with deionized water and dried overnight after being filtered using Whatman filter paper No. 41.

### 2.4. Synthesis of ZnO and MgO Nanoparticles

The precipitation method was used to synthesize the ZnO or MgO nanoparticles by dissolving 2.7 g or 4.9 g of zinc chloride or magnesium sulfate in 100 mL of deionized water to form a 0.2 mol L^−1^ concentration. A 0.02 mol L^−1^ aqueous sodium hydroxide solution was produced and added dropwise with constant agitation until the pH reached 14. The solution was kept aside until the complete precipitation was formed, then filtrated, and neutralized by washing three times with absolute ethanol. The precipitate was then dried at 80 C for 12 h before being calcined at 500 °C for 4 h. [71].

### 2.5. Sensor Construction and Membrane Composition

A traditional (AT-PT) coated wire sensor was made from a mixture of electroactive material (AT-PT, 10 mg), plasticizer (*o*-NPOE, 0.35 mL in PVC, 190 mg), and THF (5 mL). The resulting polymeric slurry was placed into a Petri dish and allowed to gently evaporate at room temperature. Deionized water, followed by acetone, was used to polish and clean the aluminum wire’s tip. The cleaned wire’s tip was dipped in the polymeric membrane solution (AT-PT) multiple times until a coated membrane formed on its surface. A thin layer membrane was formed on the surface of the modified sensor by dipping it three times in a polymeric solution containing ZnONPs or MgONPs, in addition to the previous conventional composition. The sensor was left to dry before being dipped in the aforesaid polymeric membrane formulation multiple times until a uniform coating was achieved. The Al wire/coated membrane/test solution/Ag/AgCl reference electrode was used in both coated wire sensors. The potentiometric system and the modified AT-PT-ZnONPs and AT-PT-MgONPs sensors were constructed as shown in Figure 2.

### 2.6. Calibration Procedure

The electrodes (indicator and reference) were inserted into an electrochemical cell with varying concentrations of atropine ranging from 1.0 × 10^−3^ to 1.0 × 10^−8^ mol L^−1^ to calibrate the conventional coated wire or modified sensors. The potential (E, mV) was recorded vs. the atropine concentration once the potential had stabilized for a specific concentration. By graphing the potential (E, mV) against −log [AT], a calibration curve was created, and the resulting calibration graph was utilized to quantify the unknown atropine concentrations.

### 2.7. Atropine Dosage Form Determination

In a 50 mL measuring flask, five atropine injectable ampules or eye drops were mixed thoroughly with a reasonable amount of water, agitated thoroughly, and the capacity was made up with water. A suitable volume of the resulting solution was diluted into a new 50 mL measuring flask, which was then filled to the mark with distilled water. The electrochemical cell was filled with an adequate volume of the prior solution. The potential (E, mV) of the test solution’s proposed sensors was measured versus the atropine concentration. The concentration was calculated using the calibration curve that had previously been created.

## 3. Results and Discussion

### 3.1. Characterization of the Synthesized ZnONPs and MgONPs

The size of the nanoparticles has a remarkable effect on the overall features of the materials. Understanding the material properties requires a thorough understanding of the size history of semiconducting nanoparticles. The method of UV–Visible absorption spectroscopy was widely used to study the optical properties of the nanoscale particles. The optical characteristics of each generated metal oxide nanoparticle were assessed using UV–Vis spectroscopy. A significant signal attributed to the ZnONP surface plasmon resonance (SPR) was seen at 347 nm (Figure 3a). The absorption spectra of the MgONPs revealed a single peak at 295 nm (Figure 3b). The observed spectra were found to be consistent with those published in the literature [72,73]. ZnONPs and MgONPs have anticipated bandgaps of 3.40 and 3.98 eV, respectively. A red shift in the absorption spectra of ZnONPs and MgONPs to 347 and 295 nm, respectively, and a drop in bandgap values showed the presence of nanostructured materials.

The possible functional groups that can be present in the produced nanoparticles were confirmed using FTIR analysis. The FTIR spectrum of the ZnONPs revealed a series of absorption bands ranging from 500 to 4500 cm^−1^. The 3500 nm absorption band corresponded to the water’s O–H stretching vibration. At 2370 cm^−1^, the absorption band corresponded to a strong O = C = O carbon dioxide stretching vibration. Bands at 1020, 965, 720, 520, and 466 cm^−1^, which correspond to the hexagonal crystalline Zn–O nanoparticles, were observed from 1000 to 400 cm^−1^ (Figure 4a).

FTIR in the 400–4400 cm^−1^ range was used to investigate MgONPs. At 3700, 3428, 1513, 1150, 876, 564, and 447 cm^−1^, different absorption bands for MgONPs were detected. At 3700 and 3429 cm^−1^, the stretching vibration of the O–H bond was measured in two bands. CO_2_ stretching vibration as a result of ambient carbon dioxide adsorption was blamed for the weak band at 2360 cm^−1^. The existence of an O–H stretching mode of water was shown by two newly found absorption bands about 1512 and 1400 cm^−1^. The high peak of magnesium hydroxide (Mg–OH) was discovered at 876 cm^−1^. The detected peaks, which appeared from 568 to 447 cm^−1^, confirmed the creation of Mg–O stretching vibrations (Figure 4b).

Figure 5a displays the X-ray diffraction (XRD) pattern of the generated ZnONPs. The values were seen at the 31.84° (1 0 0), 34.52° (0 0 2), 36.38° (1 0 1), 47.64° (1 0 2), 56.7° (1 1 0), 63.06° (1 0 3), and 69.18° (1 1 2) planes, and were collected between the ranges of 30–80. All of the detected peaks were hexagonal crystalline and matched the structure of zinc oxide wurtzite (JCPDS Data Card No: 36–1451) [74]. It was further proven that the produced ZnONPs were devoid of contaminants because no XRD peaks other than zinc oxide peaks were present. Figure 5b displays the XRD pattern of the MgO nanoparticles that were produced. Peaks were seen at 32.60 (1 1 1), 37.2° (2 0 0), 51.4° (2 2 0), 59.5° (2 2 1), 62.3° (3 1 1), 69.8° (2 2 2), and 72.4° (3 2 1). The acquired data revealed that the nanoparticles had a rod-like structure, which was consistent with JCPDS card number 76–1363 [75]. The size of the particle size drops as the width of the peak grows, which is similar to the current material in the nano range.

The Scherrer formula was applied to estimate the average crystallite size using the dominant peak in the XRD pattern.
D = 0.9λ/ β cos θ(1)
where the X-ray wavelength, Bragg diffraction angle, and full width at half maximum of the XRD peak occurring at a diffraction angle were, respectively, λ, β, and θ of the XRD peak occurring in the pre-synthesized ZnONPs and MgONPs had crystallite sizes of 17.9 ± 1.2 and 25.6 ± 2.5, respectively. SEM coupled with EDX was used to visualize the surface shape and elemental presence in the pre-synthesized ZnONPs and MgONPs. The SEM images picked at different magnifications revealed that the majority of the synthesized ZnONPs (Figure 6a,b) and MgONPs (Figure 6c,d) were cubic fluorite and rod-like in shape, respectively.

The EDX analysis and elemental mapping were performed and the outcomes revealed the presence of Zn with a weight % of 57.39% and atomic % of 25.10%, and O with a weight% of 16.21% and atomic % of 28.97%. Meanwhile, the EDX spectrum of MgONPs showed the presence of Mg with a weight % of 8.91% and atomic % of 6.53%, and O with a weight % of 41.03% and atomic % of 45.67% (Figure 7a,b).

### 3.2. Characteristics of Fabricated Atropine Sensors

According to the IUPAC criteria [76], the analytical categorization of atropine sensors was performed using traditional, modified sensors (ZnO, and MgO), *o*-NPOE (as a plasticizer), and PVC (as matrix). The analytical characteristics of the proposed methods are presented in Table 1. The linear equations of the calibration graphs can be expressed as follows: where *E* is the potential of the sensor (mV) and S is the slope (52 ± 0.5, 56 ± 0.5, and 54 ± 0.5 mV/decade, respectively). The intercept of the fabricated sensors was 222.3 ± 0.5, −294.1 ± 0.5, and −422.8 ± 0.5 for conventional AT-PT, modified AT-PT-ZnONPs, and AT-PT-MgONPs, respectively.
(2)$$E=E_{o}+Slog\left[atropine\right]$$

Modification of the membrane with high surface area to volume ratio metal oxides and new exceptional physicochemical properties improved the electrical conductivity of the modified sensor toward the detection of the tested analyte in the sample. Moreover, when nanomaterials were being used as transducing materials in sensing applications, the remarkable electrical and exceptional capacity properties of metal oxides such as strong charge transfer at nanomaterial interfaces are crucial [77].

In comparison to normal sensors, those improved with metal oxides have faster response times and greater mechanical stability. The functionalization of the membrane with metal oxides such as ZnONPs or MgONPs (high surface area to volume ratio) and their new advanced properties boost the electrical conductivity of the modified sensor toward the detection of AT in the sample. Furthermore, exceptional electrical capacity qualities and charge transport at nanomaterial interfaces are crucial for using nanoparticles as conductive materials in sensing systems. The selection of nanostructured materials and the sensor design approach is crucial for creating ultrasensitive sensors with the necessary properties. The size and form of the used nanoparticles define the surface-to-volume ratio, which is important for boosting the contact reactions on the overall electrical conductivity of the nanomaterials. As a result of the high chemical stability of these nanomaterials, the nanoscale shape will have an impact on the sensor’s sensitivity as well as its dynamic responsiveness and long-term stability. The electrical conductivity of sensors made from metal oxide nanomaterials may be influenced by the molecular structure and polymeric media such as crystallinity and long-chain polymers [77].

### 3.3. Response Mechanism of the Proposed Sensors

The mechanism response of the polymeric sensor based on the ion–associate complex is dependent on the equilibrium or exchange of the analyte process at the polymeric interface. The polymeric sensor frequently contains hydrophobically trapped, mobile sites. The principal mechanism of the polymeric sensors can be introduced by using simple concepts of ion–transfer equilibrium at polymeric interfaces. Such membranes with charged sites are named sited membranes. Ions of the opposite sign in the membrane are counter ions. Ions of the same sign as the sites not present in significant quantities are known as coions. Sited membranes are selective to counter ions (i.e., only counter ions exchange into the membrane and therefore have some mobility in the membrane bulk) [78,79,80].

The ion exchange process of an analyte ion (A) displaces a cation (K) from the lipophilic R- in a membrane according to the following reaction:KR (mem) +A + (aq) ⇋ AR(mem) + K^+^ (aq)

The use of modified metal oxide nanomaterials in construction sensors has been proposed to improve the sensitivity and selectivity of the proposed sensor. The improved sensitivity of the sensor was carried out by increasing the surface area to volume ratio. On the other hand, it increased the electroconductivity of the sensor, where both factors improved the limit of quantification and the detection range of the tested drug with good accuracy and precision [81,82].

### 3.4. Effect of pH

Hydrogen ion concentration greatly affects the sensor response. and therefore, it is very necessary to study the hydrogen concentration on the electrode response ability. The effect of pH is graphically illustrated in Figure 8a,b. As presented in Figure 8, the slope (E, mV) of the developed sensors (per 10-fold change) was constant with 52, 56, or 54 mV for the conventional, modified AT-PT-ZnO, and AT-PT-MgO sensors, respectively, at pH 4–9. In an alkaline medium of pH more than 9 (pKa 9.84) [83], the concentration of un-protonated atropine increases, and as a result, the potential decreases.

At an acidic medium of pH 4, limited reactivity of the protonated ion–pair complex of the sensor to AT ions was observed due to the high [H^+^] in the solution. In contrast, in an alkaline medium (pH > 9) with high [OH^−^], the potential readings gradually decreased as a result of the competition between the AT ions and OH^−^ ions. As a result, there were less interactions between the drug ions under investigation and the ion–pair sites on the sensor membrane.

The time required for the electrode to reach the steady-state potential (E, mV), is defined as the response time. It is an important factor in the characterization of sensors. The response time at a higher concentration of atropine was shorter compared with the low concentration of atropine. The average response time was 20 s for the atropine sensors. The lifetime of the membranes was defined as the time interval between the sensors’ fabrication and the modification of one characteristic response parameter. The sensors had a life limit of more than 50 days, during which time their analytical characterization remained unaltered.

### 3.5. Effect of Interferents

A separate solution and mixed solution method [84,85] were applied to evaluate the interference effect of some interfering ions. The modified sensors (ZnONPs) and magnesium oxide (MgONPs) exhibited good selectivity. The selectivity coefficient data of the proposed sensors are accessible in Table 2. The results indicate the high selectivity that was expressed by a low selectivity coefficient. The selectivity coefficients calculated using the separate solution approach were calculated as follows:(3)$$\log K_{A,\;B}^{pot}=\;\;\frac{E_{B-}E_{A}}{S}+\left[1-\frac{Z_{A}}{Z_{B}}\;\;\right]\log a_{A}$$
where (log $K_{A,\;B}^{pot}$), *E_A_*, and *E_B_* are the selectivity coefficients, potentials, and interfering species (equal concentration) of the examined sensors, respectively. While *a_A_* represents atropine activity, *Z_A_* and *Z_B_* represent the atropine charges and interfering species, respectively. Meanwhile, the equation was used to calculate the selectivity coefficient determined by the mixed solution approach.
(4)$$K_{A,B}^{pot}=\frac{\left({a^{\prime }}_{A}-a_{A}\right)}{a_{B}}$$
where *a′_A_* is the known atropine concentration added to an unknown atropine concentration *a_A_*. The potential change (*E*) was recorded. A solution with a known concentration of interfering ion (*A_B_*) was added to a predetermined concentration of atropine until the same potential was attained in another test experiment. Table 2 showed the results of the investigated sensors’ selectivity coefficients. The strong selectivity of the suggested sensors is indicated by the low selectivity coefficient values.

### 3.6. Method Validation

#### 3.6.1. The Limit of Quantification (LOQ) and Detection (LOD)

The international union of pure and applied chemistry [76] recommendation was employed to ensure the validity and suitability of the developed method. The calibration curve was logarithmic, as presented by the next equation:(5)X=Slog [atropine]+y 
where *X* represents the potential (E, mV); S represents the slope; and *y* represents the intercept. For traditional AT-PT, modified AT-PT-ZnONPs, and AT-PT-MgONPs, the calibration ranges were 1.0 × 10^−6^ to 4.0 × 10^−2^, 6 × 10^−8^ to 1 × 10^−3^, and 7 × 10^−8^ to 1 × 10^−3^ mol L^−1^, respectively, at pH 4–9. The LOD of atropine was determined based on the concentration of atropine related to the intersection of the extrapolation lines of the calibration graph, according to the IUPAC recommendations [76], with a LOQ = 3.3 LOD. For the above-mentioned sensors, LOD was 2.0 × 10^−6^, 1.8 × 10^−8^, and 2.5 × 10^−8^, whilst LOQ was 8.0 × 10^−6^, 7.0 × 10^−8^, and 8.0 × 10^−8^, respectively.

#### 3.6.2. Accuracy and Precision

The accuracy of the investigated potentiometric method was studied using seven and nine atropine concentrations, and the results were expressed as the mean percentage recoveries of 97.8, 97.8 ± 2.8, 97.30 ± 2.83, and 97.70 ± 2.66 for the conventional and modified sensors of AT-PT-ZnONP and AT-PT-MgONP, respectively. On the other hand, the precision of the proposed sensors was calculated in terms of intra- and inter-day, which was 2.8%, 2.83%, and 2.66% for the same sequence, respectively. The results are presented in Table 3.

#### 3.6.3. Ruggedness

The ruggedness of the investigated sensors using two different operators and two different instruments was expressed by good accuracy and precision. The recorded RSD % on the day- and inter-day was less than 3.2%.

### 3.7. Repeatability, Reproducibility, and Stability of the Modified Sensors

The repeatability study for the AT-PT/ZnONP and AT-PT/MgONP sensors was recorded by constructing two calibration curves through six successive days, using each sensor separately, with AT sample concentrations of 6.0 × 10^−8^ − 1.0 × 10^−3^, and 8.0 × 10^−8^ − 1.0 × 10^−3^ mol L^−1^. The obtained results showed that the obtained slopes were 55.8 ± 0.2 and 53.6 ± 0.7 with lower limits of detection of 2.0 × 10^−9^ and 1.0 × 10^−9^ mol L^−1^ for the above-mentioned sensors, respectively. The obtained results revealed the high repeatability of the suggested modified sensors. The slopes corresponding to the modified sensors were 55.8 ± 0.2 and 53.6 ± 0.7 mV/decade, respectively. This supports the adequate reproducibility of this sort of sensor. This revealed that the proposed modified sensors can be used adequately at least for 6 consecutive months.

A TGA analysis was performed to examine the thermal stability of the modified metal oxide membranes. This technique can be used to examine the mass loss of a substance caused by the oxidation, reduction, or the loss of volatile substances such as moisture that are present within a fixed range of temperature, which typically yields a temperature (or time) versus mass (or weight percentage) plot [86]. The thermal deterioration of ZnONPs and MgONPs in the modified membranes was assessed in the temperature of 30–600 °C. The dehydration of the surface-adsorbed water causes a weight loss between room temperature, 30 °C, and 100 °C, and it was noticed that only 1.8% and 2.4% weight loss were recorded for the modified membranes with ZnONPs and MgONPs, respectively. No further significant wight loss was observed at higher temperature, revealing the excellent stability of the fabricated membranes (Figure 9).

### 3.8. Application of Atropine Sensors

The developed sensors were used for the atropine assay in pure solutions and pharmaceutical formulations. The determination of atropine (*n* = 5) was recorded using the conventional and modified sensors. The accuracy was 98.5%, 99.1%, and 98.6% while the precision was 1.4%, 3.1%, and 2.8% for the conventional, modified ZnONP, and MgONP sensors, respectively (Table 4). The created ZnONP and MgONP sensors were used to quantify atropine in its dose form, with the typical recovery of 98.0%, and 98.3%, respectively, with the precision of 2.3%, and 2.5% for the modified sensors ZnONPs and MgONPs, respectively (Table 5). The results of the atropine assay in its pharmaceutical form were compared to those obtained using the British pharmacopeia approach (Table 5) [87]. Statistical analysis of the created sensors and the described spectrophotometric method for the atropine assay indicated no significant differences in the accuracy and precision between the developed sensors and the published method using the null hypothesis method with a 0.05 cutoff and *n* = 5 subjects (Table 5) [88].

The efficiency of the constructed functionalized AT-PT-ZnONP and AT-PT-MgONPs sensor was compared to that of the previously reported sensors [89,90,91] (Table 6).

## 4. Conclusions

The current study describes simple and ultrasensitive modified AT-PT-ZnOMP and AP-PT-MgONP potentiometric sensors to determine the AT in authentic powder and commercial formulations that was successfully fabricated. The improved sensor was constructed with a large surface area to volume ratio, which allowed for exceptional sensitivity in the detection of AT with linear relationships in the concentration ranges of 6.0 × 10^−8^ − 1.0 × 10^−3^ and 8.0 × 10^−8^ − 1.0 × 10^−3^ mol L^−1^ with regression equations of E_(mV)_ = (56 ± 0.5) log [AT] − 294 and E_(mV)_ = (54 ± 0.5) log [AT] − 422 for the AT-PT-NP or AT-PT-MgONP sensors, respectively. The results of the proposed potentiometric systems were statistically assessed and compared to those of the official method. It was revealed that the modified sensors had a higher potential response than the conventional type. Furthermore, covering the sensor’s surface with a modified layer of metal oxide nanoparticles improved the sensor’s electroconductivity and the quantification of the tested drug in tablets, with a good recovery percentage, indicating high sensitivity and selectivity. As a result, using metal oxide nanoparticles in the construction of polymeric sensors opens up a new avenue for producing unique modified potentiometric sensors.

## Figures and Tables

**Figure 1 nanomaterials-12-02313-f001:**
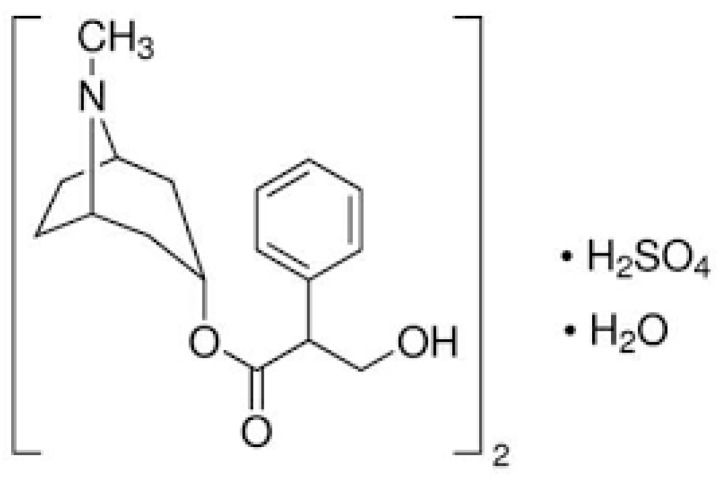
The chemical structure of atropine sulfate.

**Figure 2 nanomaterials-12-02313-f002:**
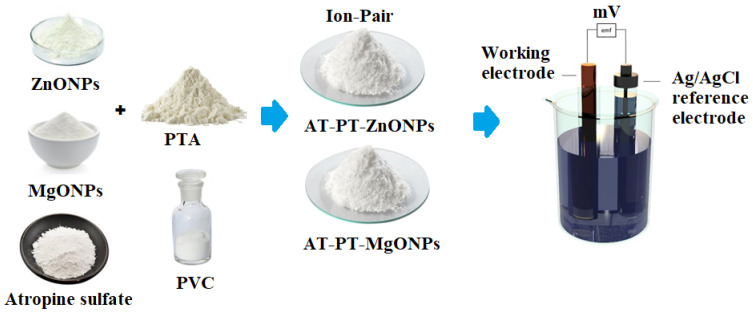
The preparation of modified AT-PT-ZnONP or AT-PT-MgONP sensors and the potentiometric system.

**Figure 3 nanomaterials-12-02313-f003:**
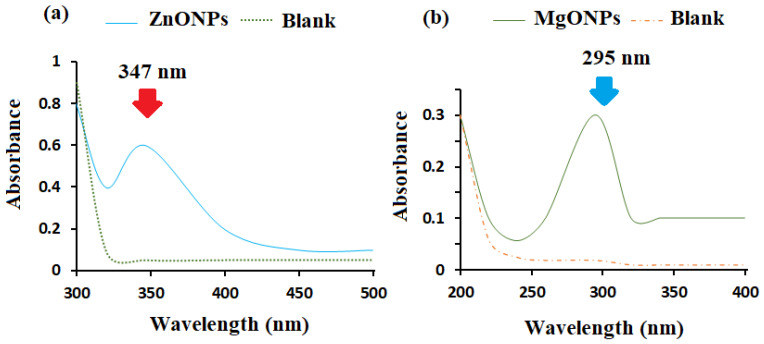
The UV–Vis spectra of (**a**) ZnONPs and (**b**) MgONPs at 347 and 295 nm, respectively.

**Figure 4 nanomaterials-12-02313-f004:**
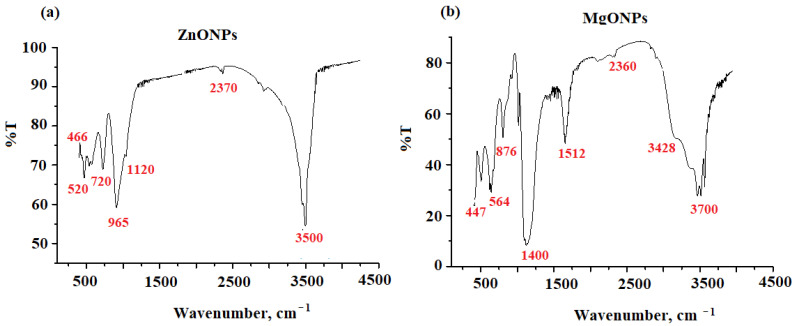
The FTIR spectra of the (**a**) ZnONPs and (**b**) MgONPs measured at wavenumber in the range of 500–4500 cm^−1^.

**Figure 5 nanomaterials-12-02313-f005:**
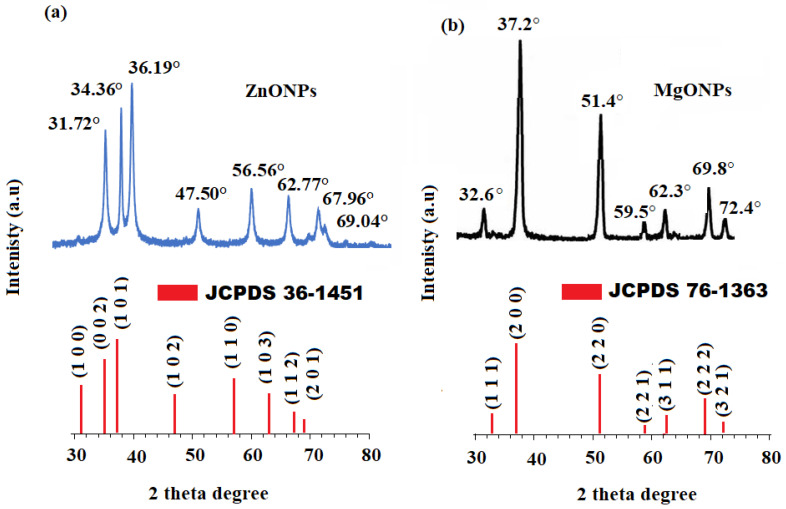
The XRD patterns of the (**a**) ZnONPs and (**b**) MgONPs detected in the range 30–80 theta degrees.

**Figure 6 nanomaterials-12-02313-f006:**
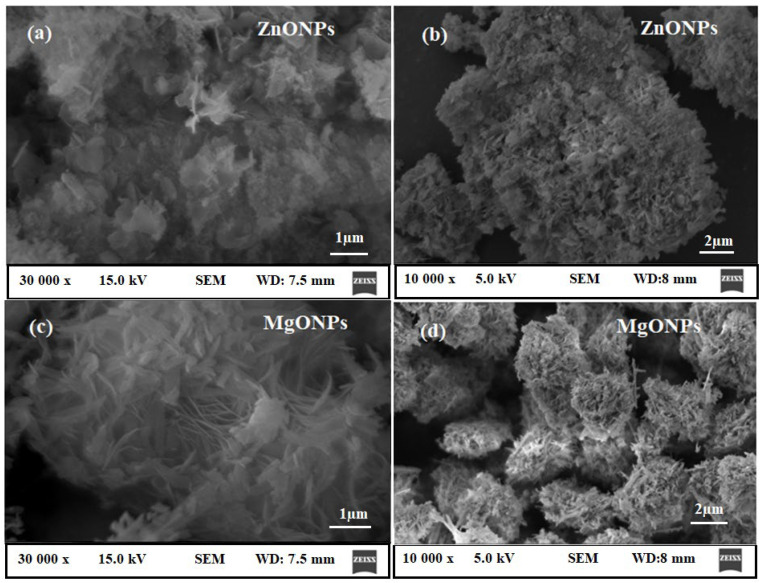
The SEM images of the (**a**,**b**) ZnONPs and (**c**,**d**) MgONPs measured at different magnifications.

**Figure 7 nanomaterials-12-02313-f007:**
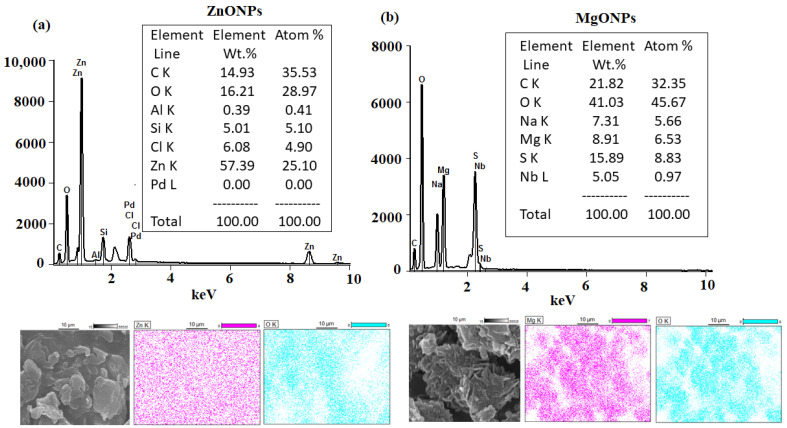
The EDX spectra and mapping analysis of the (**a**) ZnONPs and (**b**) MgONPs.

**Figure 8 nanomaterials-12-02313-f008:**
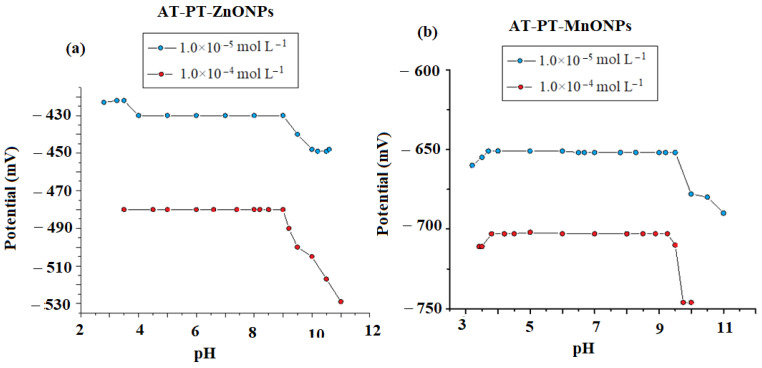
The effect of pH on the investigated (**a**) AT-PT-ZnONP and (**b**) AT-PT-ZnONP sensors using two AT concentrations of 1.0 × 10^−5^ and 1.0 × 10^−4^ mol L^−1^.

**Figure 9 nanomaterials-12-02313-f009:**
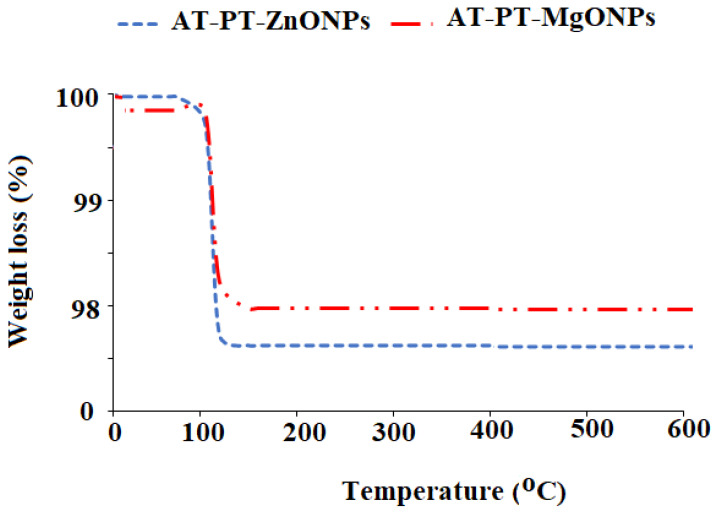
The thermal stability of the modified AT-PT-ZnONP and AT-PT-MgONP membranes measured by the TGA analysis.

**Table 1 nanomaterials-12-02313-t001:** The analytical characterization of the designed conventional coated wire AT-PT and modified (AT-PT-ZnONPs and AT-PT-MgONPs) sensors.

Parameter	Conventional Coated Wire (AT-PT)	Modified (AT-PT-ZnONPs)	Modified (AT-PT-MgONPs)
Slope, (mV decade^−1^)	52	56.1	55
Intercept, mV	198 ± 0.5	−294.1 ± 0.5	−422.8 ± 0.5
Correlation Coefficient, (r)	0.998	0.999	0.999
Calibration, rang mol L^−1^	4.0 × 10^−6^ − 1 × 10^−3^	6.0 × 10^−8^ − 1.0 × 10^−3^	8.0 × 10^−8^ − 1.0 × 10^−3^
LOQ, mol L^−1^	4.0 × 10^−6^	6.0 × 10^−8^	8.0 × 10^−8^
LOD, mol L^−1^	2 × 10^−6^	1.8 × 10^−8^	2.5 × 10^−8^
Response time, s	20	20	20
Working pH range	4–9	4–9	4–9

**Table 2 nanomaterials-12-02313-t002:** The selectivity coefficients of the proposed AT-PT, AT-PT-ZnONP, and AT-PT-MgONP coated wire sensors.

Interferent, J *	$K_{AT,\;B\;}^{pot}$ AT-PT	$K_{AT,\;B\;}^{pot}$ AT-PT-ZnONPs	$K_{AT,\;B\;}^{pot}$ AT-PT-MgONPs
Na^+^	1.7 × 10^−2^	1.6 × 10^−4^	5.3 × 10^−3^
K^+^	1.5 × 10^−2^	3.5 × 10^−4^	5.3 × 10^−3^
Magnesium Stearate	1.2 × 10^−2^	2.0 × 10^−4^	5.1 × 10^−3^
Acetate	1.4 × 10^−2^	1.8 × 10^−4^	5.1 × 10^−3^
Citrate	1.3 × 10^−2^	6.6 × 10^−4^	5.1 × 10^−3^
L- Tryptophan *	2.2 × 10^−3^	6.5 × 10^−4^	5.6 × 10^−3^
DL-Alanine *	1.1 × 10^−3^	4.5 × 10^−4^	5.7 × 10^−3^
Glycine *	1.2 × 10^−3^	6.6 × 10^−4^	5.6 × 10^−3^
Glucose *	1.3 × 10^−3^	6.3 × 10^−4^	5.7 × 10^−3^
Lactose monohydrate *	2.1 × 10^−3^	6.4 × 10^−4^	5.6 × 10^−3^
Starch *	2.1 × 10^−3^	6.6 × 10^−4^	5.7 × 10^−3^
Microcrystalline cellulose *	2.1 × 10^−3^	6.6 × 10^−4^	5.7 × 10^−3^

* Match potential method.

**Table 3 nanomaterials-12-02313-t003:** The accuracy study for the AT estimation using the designed conventional coated wire AT-PT and modified AT-PT-ZnO and AT-PT-MgO nanocomposite sensors.

Sample	Conventional AT-PT		Modified AT-PT-ZnONPs		Modified AT-PT-MgONPs	
	−log [AT], mol L^−1^	Recovery %	−log [AT], mol L^−1^	Recovery %	−log [AT], mol L^−1^	Recovery %
Statistical Analysis			7	97.0	7	97.0
	6	97.0	6.5	97.0	6.5	97.0
	5.5	97.5	6	97.5	6	97.0
	5	97.5	5.5	97.5	5.5	98.0
	4.5	98.0	5	97.5	5	98.0
	4	98.0	4.5	97.5	4.5	98.0
	3.5	98.0	4	98.0	4	98.0
	3	99.0	3.5	98.0	3.5	98.0
			3	99.0	3	98.5
Mean ± SD	97.8 ± 2.8		97.30 ± 2.83		97.70 ± 2.66	
n	7		9		9	
RSD, %	2.8		2.8		2.7	

**Table 4 nanomaterials-12-02313-t004:** The intermediate precision of the design of the AT-PT-ZnO and AT-PT-MgO nanocomposite sensors.

Sample	Modified AT-PT-ZnONPs			Modified AT-PT-MgONPs		
	Intra-Day		Inter-Day	Intra-Day		Inter-Day
	−log [AT], mol L^−1^	Recovery %	Recovery %	−log [AT], mol L^−1^	Recovery %	Recovery %
Statistical Analysis	7	97.0 ± 2.9	97.0 ± 2.8	7	97.0 ± 2.8	97.0 ± 2.9
	6.5	97.5 ± 2.9	97.5 ± 2.8	6.5	97.5 ± 2.8	97.5 ± 2.9
	5.5	97.5 ± 2.7	97.5 ± 2.8	5.5	97.5 ± 2.7	97.0 ± 2.7
	4.5	98.0 ± 2.7	97.5 ± 2.7	4.5	98.0 ± 2.7	98.0 ± 2.7
	3.5	98.0 ± 2.5	98.0 ± 2.7	3.5	98.5 ± 2.5	98.0 ± 2.6
	3.0	99.0 ±2.5	98.0 ± 2.6	3.0	99.0 ± 2.5	98.5 ± 2.6
Mean ±SD		98.00 ± 2.7	97.64 ± 2.7		97.91 ± 2.6	97.6 ± 2.7
n		6	6		6	6
RSD, %		2.7	2.7		2.7	2.7

**Table 5 nanomaterials-12-02313-t005:** The calculated results of atropine using the conventional coated wire AT-PT and modified AT-PT-ZnONP and AT-PT-MgONP sensors.

Sample	Conventional AT-PT	Modified AT-PT-ZnONPs	Modified AT-PT-MgONPs	Reported Method [89]
	Recovery ± RSD	Recovery ± RSD	Recovery ± RSD	Recovery± RSD
Atropine injection (1 mg/mL)	97.5 ± 2.7	97.4 ± 2.8	97.5 ± 2.8	97.5 ± 2.80
Atropine Eye drop (1%)	97.3 ± 2.9	97.5 ± 2.8	97.4 ± 2.7	98.4 ± 2.60
t-test F test	0.75 (3.36) * 0.20 (6.38) *	0.00 (3.36) * 0.84 (6.38) *	0.02 (3.36) * 0.99 (6.38) *	

* Represents the tabulated values of t and F.

**Table 6 nanomaterials-12-02313-t006:** The comparative outcomes between the suggested modified AT-PT-ZnONP and AT-PT-MgONP sensors and the previously reported sensors.

No.	Ion-Pair Complex	Linear Concentration Range (mol L^−1^)	LOD (mol L^−1^)	References
1.	AT-tetraphenylborate	1.0 × 10^−5^ −1.0 × 10^−2^	3.29 × 10^−6^	[89]
2.	AT-tetraphenylborate	1.0 × 10^−5^ −1.0 × 10^−2^	2.0 × 10^−5^	[90]
3.	AT-Tetrakis tri-methylfluoro phenyl borate	1.0 × 10^−6^ − 9.1 × 10^−3^		[91]
4.	AT-PT-ZnONPs AT-PT-MgONPs	6.0 × 10^−8^ − 1.0 × 10^−3^ 8.0 × 10^−8^ − 1.0 × 10^−3^	1.8 × 10^−8^ 2.5 × 10^−8^	Current study

## Data Availability

The outcome of data from this study.

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
