# Peer review of "Atropine-Phosphotungestate Polymeric-Based Metal Oxide Nanoparticles for Potentiometric Detection in Pharmaceutical Dosage Forms"

_nanomaterials, 2022, doi:10.3390/nano12132313_

Round 1

Reviewer 1 Report

This manuscript entitled: “Atropine-Phosphotungestate Polymeric-based Metal Oxide Na- noparticles for Potentiometric Detection in Pharmaceutical Dosage Forms” developed suitable nanomaterials and used them for atropine analysis. The results are solid. I think it is suitable for publication in Nanomaterials with some minor revisions.

1.      The effective numbers of Line 341, and 342 need to be double checked to be consistent with other numbers.

2.      How about the stability of the nano-materials’-based electrodes? Can they be repeatedly used for measurement?

Author Response

Nanomaterials

Manuscript ID: nanomaterials-1789677

Comments to Reviewer 1

  1. The effective numbers of Line 341, and 342 need to be double checked to be consistent with other numbers.

Answer: After revision of the text the lines will be 345 and 346 and the numbers have been corrected.

  1. How about the stability of the nano-materials’-based electrodes? Can they be repeatedly used for measurement?

          Answer: The repeatability and reproducibility and stability of the modified sensors have been cited in the revised text.

Reviewer 2 Report

The authors designed atropine-phosphotungestate polymeric-based metal oxide nanoparticles for potentiometric detection in pharmaceutical dosage forms. Although being interesting, I find that there are some major issues with the paper that require addressing prior to this being considered for publication in this journal. I have identified the main points for consideration below:

1.     This manuscript has some spelling typos, style errors and grammatical errors. Pleases carefully check and correct them in the revised manuscript.

2.     The sensing principle of the proposed sensor should be classified in the revised manuscript, and a scheme of sensing principle is also needed.

3.     The repeatability, reproducibility and stability of this sensor should be investigated in the revised manuscript.

4.     In the introduction section, the advantages of electrochemical sensors should be mentioned and some recent references are recommended to be cited, including Journal of Hazardous Materials 436 (2022) 129107; Microchemical Journal 179 (2022) 107515.

5.     The quality of the figures should be improved.

6.     The sensing performance of this sensor should be compared with the previously reported ones.

7.     What role did ZnONPs or MgONPs play?

8.     The effect of pH on response potential should be discussed in the revised manuscript.

Author Response

Nanomaterials

Manuscript ID: nanomaterials-1789677

Comments to Reviewer 2

  1. This manuscript has some spelling typos, style errors and grammatical errors. Pleases carefully check and correct them in the revised manuscript.

Answer: The manuscript has been carefully checked, all spelling typos, style errors and grammatical errors have been corrected in the revised text. 

2.The sensing principle of the proposed sensor should be classified in the revised manuscript, and a scheme of sensing principle is also needed.

Answer: The sensing principle of the proposed sensor with scheme has been added in the text.

  1. The repeatability, reproducibility and stability of this sensor should be investigated in the revised manuscript.

Answer: The repeatability, reducibility, and thermal stability of the modified sensors have been cited in the revised text.

  1. In the introduction section, the advantages of electrochemical sensors should be mentioned and some recent references are recommended to be cited, including Journal of Hazardous Materials 436 (2022) 129107; Microchemical Journal 179 (2022) 107515.

Answer: Introduction section has been improved and recent references have been included and also the recommended references have been cited.

  1. The quality of the figures should be improved.

Answer: The quality of Figures has been improved

  1. The sensing performance of this sensor should be compared with the previously reported ones.

Answer: The sensing performance of the modified sensors has been compared with previously reported sensors (Table 6).

  1. What role did ZnONPs or MgONPs play?

Answer:  The enhancement effect of metal oxides on the modified sensors has been added in the text (lines 297-311). 

  1. The effect of pH on response potential should be discussed in the revised manuscript.

Answer: The effect of pH on potential response has been discussed in the revised text.

Round 2

Reviewer 2 Report

 Accept in present form.